# Effects of similarity networks in graph-based multi-omics classification

**Masrafe Bin Hannan Siam** , **Md Rayhan Khan**, **Md Fazla Elahe** *, **Md Shohel Arman**, **Swarna Akter**

Data Science Lab, Department of Software Engineering, Daffodil International University, Daffodil Smart City, Birulia, Dhaka, Bangladesh

◉ These authors contributed equally to this work.
* elahe.se@daffodilvarsity.edu.bd

## Abstract

Accurate classification of disease subtypes is a fundamental requirement of precision medicine especially for complex and heterogeneous conditions such as breast cancer and Alzheimer's disease. Recent advances in graph-based deep learning have shown strong potential in multi-omics integration by modeling inter-sample relationships through similarity networks. Yet, the question of how best to construct these networks remains an open and underexplored challenge. In this work, we present a systematic evaluation of six distinct similarity network construction strategies including Cosine Similarity, Cosine Distance, RBF-based measures, and two hybrid combinations leveraging a graph convolutional network (GCN) integrated with a view correlation discovery network (VCDN) framework for multi-omics disease classification. Using two benchmark datasets (BRCA and ROSMAP), we assessed the impact of each method on classification performance, variance across runs, and statistical robustness. Surprisingly, our results demonstrate that Cosine Similarity outperforms all other metrics, consistently achieving the highest accuracy, F1-score, and AUC, while also showing the lowest standard deviation across cross-validation splits. Despite the growing popularity of kernel-based and hybrid similarity designs, our findings highlight the unique effectiveness of simple angular similarity in capturing biologically meaningful structure in high-dimensional omics data. In our study, we showed that simple yet biologically meaningful similarity measures like Cosine Similarity can outperform more complex techniques in accuracy, consistency, and clarity. This insight sets the stage for building more effective and interpretable graph-based models to support precision medicine.

## Introduction

Cancer encompasses a wide spectrum of diseases that can affect nearly any organ or tissue in the human body [1]. A major objective in cancer research is the

**Data availability statement:** All relevant data are within the manuscript and Supporting Information files.

**Funding:** The author(s) received no specific funding for this work.

**Competing interests:** The authors have declared that no competing interests exist.

identification of disease subtypes and the evaluation of patient prognosis [2]. Traditional diagnostic approaches often fall short in capturing the biological complexity and heterogeneity characteristic of diseases like cancer. The advent of high-throughput sequencing technologies has facilitated the collection of large-scale datasets across multiple omics layers, including microRNA (miRNA) expression, DNA methylation, and mRNA expression. Numerous studies have shown that integrating multi-omics data leads to more accurate disease prediction than relying on single-omics data alone [2–4].

Several deep learning frameworks have been developed to exploit the potential of multi-omics data integration. [5] For instance, SALMON is a multi-omics neural network model designed specifically for survival analysis [3]. Similarly, deep learning-based fusion of multi-omics data has improved survival prediction in liver cancer patients [4]. Attention-based graph convolutional networks have been proposed to improve disease subtype classification and biomarker discovery by modeling complex dependencies across omics modalities [6]. Hypergraph-based methods such as MORE demonstrate that integrating higher-order relationships further improves classification and biomarker detection tailored to individual patients [7]. Collectively, these studies suggest that machine learning models are increasingly effective at harnessing the complexity of multi-omics data to enhance disease subtype prediction.

A wide range of methodologies have been developed for multi-omics integration, which can broadly be categorized into unsupervised and supervised learning approaches. Unsupervised approaches are typically employed for clustering or identifying biologically meaningful groups, such as cancer subtypes, without relying on outcome labels. For example, Subtype-MGTP applies a deep subspace contrastive clustering algorithm on protein expression data to discover biologically relevant cancer subtypes [8]. However, unsupervised models often suffer from reduced precision due to the lack of labeled guidance. In contrast, supervised approaches utilize labeled data, such as clinical outcomes or disease states, to train predictive models. For instance, DIABLO is a supervised integration framework that extracts latent variables to identify key molecular drivers across omics layers. DeepCC employs a deep learning architecture to predict cancer subtypes from multi-omics profiles. [9] Likewise, an attention-based graph convolutional network for multi-omics integration has been introduced to improve predictive performance and interpretability [10]. Semi-supervised approaches, such as those incorporating transformers and graph convolutions, also show strong classification performance when labeled data is limited.

Traditional machine learning models like support vector machines and random forests treated multi-omics integration as a flat feature concatenation problem. While effective in simple scenarios, these approaches struggle with high dimensionality and fail to capture cross-omics relationships. Recent advances in deep learning have enabled more sophisticated modeling of non-linear and high-dimensional data. For instance, SALMON integrates mRNA, miRNA, and clinical data using a neural architecture [3], while other frameworks employ fully connected deep networks for prognosis prediction [4]. However, fully connected architectures typically overlook the inter-sample relationships critical to biological understanding. Graph-based learning

methods, such as DeepMoIC and MoGCN, address this limitation by integrating similarity network fusion and graph convolutions to model sample-level dependencies more effectively [11]. Attention-based GCNs have further enhanced classification accuracy by dynamically learning the importance of sample relationships [10].

Graph Convolutional Networks (GCNs) have become instrumental in representing complex sample relationships in multi-omics integration. In these models, each patient or sample is treated as a node, and edges represent similarities derived from omics data. GCNs exploit both the network structure and feature attributes to classify nodes, enabling interpretable and efficient learning. This framework has been applied to predict genome-disease associations, identify drug-disease interactions, and improve subtype classification in cancer through models like MoGCN [12] and DeepMoIC. Attention-based GCNs further extend this idea by adjusting edge weights based on feature relevance, resulting in more robust and interpretable predictions [10].

Sample similarity networks play a pivotal role in constructing effective graph-based multi-omics models. These networks model inter-sample relationships based on molecular similarity and guide graph learning. For example, residual GCNs that leverage similarity networks have been used to distinguish cancer subtypes [13]. Other models, including DeepMoIC, MoGCN [11], and MOGAT [14], use similarity graphs to enhance both prediction accuracy and biomarker identification. Adaptive attention mechanisms further refine similarity matrices during training to enhance performance [8]. Hypergraph-based integration strategies also enable modeling of higher-order sample interactions across multiple omics layers [6].

In this study, we present a graph-based multi-omics integration framework that systematically evaluates the impact of various similarity network constructions on disease subtype classification. We conduct experiments on two benchmark datasets BRCA (Breast Invasive Carcinoma) and ROSMAP (Religious Orders Study and Memory and Aging Project) to explore how similarity network design influences model performance. Our framework employs Graph Convolutional Networks (GCNs) to independently extract representations from mRNA expression, DNA methylation, and miRNA expression data. We construct omics-specific sample graphs using cosine similarity, cosine distance, radial basis function (RBF) similarity, RBF distance, and hybrid combinations. The initial outputs from each omics-specific GCN are combined into a cross-view discovery tensor, which is then processed by a View Correlation Discovery Network (VCDN) to model high-order dependencies in the label space. Through comprehensive evaluation, we demonstrate how the choice of similarity metric impacts classification accuracy and biomarker interpretability. Our findings highlight the critical role of similarity network design in enhancing the effectiveness of graph-based multi-omics integration methods for biomedical research.

## Related work

Graph-based integration methods have received significant attention in multi-omics research due to their capacity to model complex sample relationships. MOGONET (Multi-Omics Graph cOnvolutional NETworks) is a pioneering supervised framework that constructs omics-specific sample similarity graphs typically using cosine or distance-based metrics and employs Graph Convolutional Networks (GCNs) followed by a View Correlation Discovery Network (VCDN) for cross-omics fusion and classification [15,16], outperforming traditional methods in disease subtype classification and biomarker identification.

Other noteworthy graph-based architectures include MoGCN, which applies GCNs over similarity networks derived from each omics view, achieving strong performance in cancer subtype prediction [11]. DeepMoIC leverages a fusion of similarity networks via deep GCNs to improve robustness and classification accuracy. Graph attention-based models such as MOGAT introduce attention mechanisms to adaptively weight node relationships [14], while hypergraph-based methods like MORE model higher-order interactions among samples across omics layers.

Complementing these structural learning approaches, recent advancements in feature selection and bio-inspired optimization have also demonstrated significant improvements in cancer classification accuracy. Yaqoob et al. proposed efficient gene selection frameworks utilizing Brownian Motion Search with SVM [17] and hybrid approaches integrating

Cuckoo Search algorithms [18] to handle high-dimensional biological data. Similarly, Afreen et al. introduced a game-theoretic strategy combining Shapley values with improved Grey Wolf Optimization to refine feature subsets for lung and colon cancer [19]. These optimization-driven methods, along with broader AI-driven diagnostic insights [20], highlight the critical importance of preprocessing and feature quality, which works in tandem with the graph-based integration strategies discussed below.

Parallel efforts have evaluated GNN architectures across multi-omics datasets by comparing GCNs, GATs, and graph-transformer networks. These studies highlight that the quality and design of sample graph structures such as correlation- or kernel-based networks directly impact model performance [21]. For instance, correlation-based sample graphs yielded superior accuracy compared to protein-protein interaction-based graphs [21].

Despite architectural advances, fewer studies have systematically investigated how different similarity metrics influence graph construction and downstream results. Cosine similarity is favored for its robustness to scale and sparsity in omics data, while Radial Basis Function (RBF) kernels capture non-linear proximity effectively [22]. Hybrid approaches that combine cosine and RBF metrics aim to exploit both angular and spatial relationships, enhancing the expressiveness of sample graphs [10].

Even with the proliferation of graph-based architectures such as MoGCN, DeepMoIC, and MOGAT, a critical methodological gap persists: the choice of similarity metric used to construct the underlying sample graph is often treated as a fixed hyperparameter rather than a fundamental design choice. While some works favor Cosine Similarity for its scale-invariance and others employ RBF kernels to capture non-linear proximities, there has been no systematic investigation into how these choices or their hybrid counterparts influence model robustness and classification accuracy in a unified multi-omics framework. This study addresses this gap by independently evaluating six similarity network strategies within a GCN-VCDN architecture. By doing so, we move beyond simple architectural comparisons to provide practical, evidence-based guidance on the most effective ways to model inter-sample relationships in high-dimensional biological data.

Yu et al. proposed a layer-attention GCN for drug–disease prediction, which dynamically learns edge weights using combined similarity measures [23]. Similarly, Guo et al. demonstrated that models using adaptive similarity matrices outperform static graph constructions in cancer subtype classification [10]. Yet, these studies stop short of systematically comparing cosine, RBF, and hybrid network options.

In this work, we extend these efforts by systematically evaluating six similarity network strategies cosine similarity, cosine distance, RBF similarity, RBF distance, hybrid similarity, and hybrid distance within the MOGONET framework. Our analysis highlights how similarity metric choice affects classification accuracy and interpretability across BRCA and ROSMAP datasets.

## Methodology

### Graph convolutional networks (GCNs)

Graph Convolutional Networks (GCNs) are a powerful type of neural network designed to work directly with graph-structured data. This makes them especially well-suited for multi-omics integration tasks, where relationships between patients or genes play a crucial role in understanding disease mechanisms [24].

Unlike traditional Convolutional Neural Networks (CNNs), which operate on grid-like data (e.g., images), GCNs learn by aggregating information from each node's neighbors. This allows the model to capture contextual signals and relational patterns within the graph, which is essential for tasks like disease subtype prediction.

The core GCN update rule can be described as:

$$\mathbf{H}^{(l+1)} = \sigma \left( \tilde{\mathbf{D}}^{-\frac{1}{2}} \tilde{\mathbf{A}} \tilde{\mathbf{D}}^{-\frac{1}{2}} \mathbf{H}^{(l)} \Theta^{(l)} \right)$$

Here, $\tilde{A} = A + I$ adds self-loops to include each node's own information, $\tilde{D}$ is the degree matrix, $H^{(l)}$ represents node features at layer $l$, $\Theta^{(l)}$ contains trainable weights, and $\sigma$ is a non-linear activation function [24]. When stacked, multiple GCN layers allow information to flow across the graph, enabling deeper understanding of patient similarities.

For classification tasks, such as identifying disease subtypes, GCNs typically apply a softmax function on the final node embeddings and optimize the model using a cross-entropy loss. Prior works, such as AD-GCN, have shown that this approach is highly effective in capturing the complexity of heterogeneous omics data using patient similarity graphs. Their scalability, combined with the ability to incorporate biological structure, makes GCNs a promising choice for modern biomedical applications [25].

## View correlation discovery network (VCDN)

The View Correlation Discovery Network (VCDN) is a central component of MOGONET, designed to intelligently combine outputs from omics-specific GCNs by capturing dependencies between modalities at the label level [15]. Rather than simply averaging predictions, VCDN models high-order interactions between the predicted class probabilities from each omics view, allowing for a more informed and synergistic decision-making process.

For a given sample $j$, VCDN forms a cross-view tensor by computing the outer product of the predicted probability vectors $\hat{y}^{(i)} \in \mathbb{R}^c$ from all $m$ omics views:

$$C_{j,a_1 a_2 \ldots a_m} = \prod_{i=1}^{m} \hat{y}_{j,a_i}^{(i)}$$

This tensor captures all possible class combinations across views, effectively encoding the confidence and correlation structure of predictions. The tensor is then flattened into a vector $\mathbf{c}_j \in \mathbb{R}^{c^m}$, which is passed through fully connected layers and classified using a softmax function [26].

VCDN is trained jointly with the omics-specific GCNs in an end-to-end manner using cross-entropy loss. This setup allows the model to learn not only how to represent omics data within each modality, but also how different modalities interact at the decision level. Prior studies have shown that removing VCDN significantly reduces performance, underscoring its vital role in achieving robust and accurate multi-omics integration [27].

## MOGONET

MOGONET (Multi-Omics Graph cOnvolutional NETworks) is a supervised deep learning framework specifically designed to tackle disease classification challenges using multi-omics data [15]. It brings together two powerful components omics-specific Graph Convolutional Networks (GCNs) and a View Correlation Discovery Network (VCDN) to capture both within-modality structure and across-modality dependencies in a unified model.

For each omics layer (such as mRNA, miRNA, or DNA methylation), MOGONET constructs a sample similarity graph and applies a dedicated GCN. These GCNs learn feature representations that account for both individual sample profiles and their relationships to others in the same omics space. Rather than simply merging features or predictions, MOGONET uses VCDN to integrate the outputs of each GCN in a more sophisticated way by modeling how the class predictions from different omics modalities interact and correlate.

The entire system is trained end-to-end using a supervised loss, allowing both the GCNs and the VCDN to learn in sync. This approach ensures that the representations learned at the omics level are fully aligned with the label-level fusion performed across modalities.

MOGONET has shown strong classification performance on complex disease datasets like BRCA and ROSMAP. Beyond predictive accuracy, it also supports interpretability through attention mechanisms in the GCNs, making it easier

to identify important features or biomarkers [28–30]. Its flexible design also makes it easy to extend with additional omics layers or alternative similarity networks.

## Similarity network construction

To examine how similarity measures influence multi-omics graph learning, we built six types of sample similarity networks for each omics modality (mRNA, DNA methylation, and miRNA). These networks define the structure through which information flows in GCNs.

The similarity metrics include:

- **Cosine Similarity**

- **Cosine Distance**

- **RBF Similarity**

- **RBF Distance**

- **Hybrid Similarity**

- **Hybrid Distance**

These networks aim to balance global and local biological patterns, with prior work showing that similarity choice directly affects graph quality and model performance [15,31,32].

Fig 1 illustrates the full pipeline from similarity construction to model evaluation.

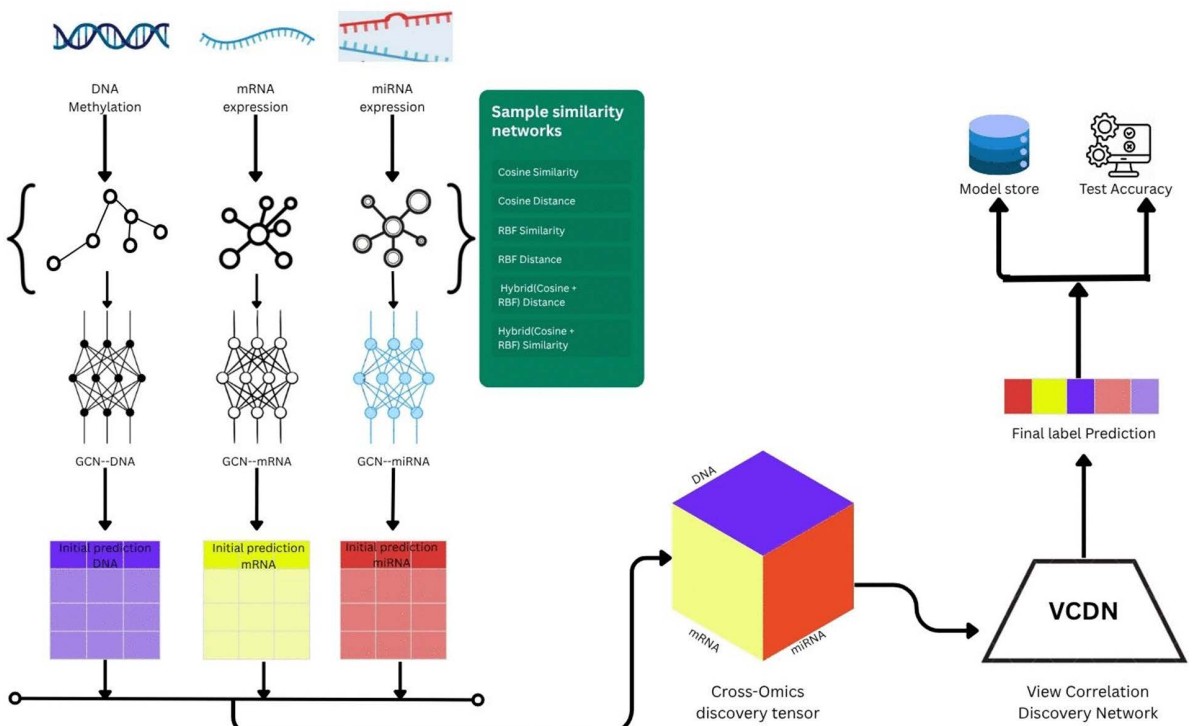

**Fig 1. Methodological pipeline for multi-omics classification.** Overview of the methodological pipeline, illustrating graph construction, model training, and evaluation phases in multi-omics classification.

## Cosine similarity

Cosine similarity is a widely used metric that captures how directionally aligned two vectors are, making it particularly effective for high-dimensional and sparse omics data. It is defined as:

$$\text{cosine\_sim}(\mathbf{x}, \mathbf{y}) = \frac{\mathbf{x} \cdot \mathbf{y}}{\|\mathbf{x}\| \, \|\mathbf{y}\|} = \frac{\sum_{i=1}^{n} x_i y_i}{\sqrt{\sum_{i=1}^{n} x_i^2} \, \sqrt{\sum_{i=1}^{n} y_i^2}},$$

where the output ranges from 0 to 1 for non-negative inputs. Because it focuses purely on vector direction and not magnitude, cosine similarity is naturally resistant to batch effects an important feature in noisy biological datasets.

In the MOGONET framework, cosine similarity is used to define edges in sample graphs. Typically, only the top-$k$ most similar samples are connected (or a similarity threshold is applied), allowing the graph to highlight biologically meaningful relationships between patients [33]. Leads to stable and interpretable graph structures that often align with known biological patterns [27]. Efficient to compute and highly robust, even in large-scale omics datasets. Ignores feature magnitudes, which may carry biological relevance in certain contexts (e.g., gene dosage). Requires careful selection of graph sparsification parameters, such as the $k$ value or similarity threshold.

Soft Cosine Similarity: Extends the basic version by accounting for correlations between features, potentially capturing more nuanced similarities.

Angular Distance: A distance counterpart defined as

$$D_{\theta}(\mathbf{x}, \mathbf{y}) = \frac{\arccos(\text{cosine\_sim}(\mathbf{x}, \mathbf{y}))}{\pi},$$

which has been used in several biologically grounded graph models.

Overall, cosine similarity remains a simple yet powerful method for building sample similarity networks in multi-omics graph learning. It strikes a useful balance between interpretability, robustness, and biological relevance [31].

## Cosine distance

Cosine distance is the complement of cosine similarity and measures how different the directions of two vectors are. It's particularly useful in high-dimensional omics data where the relative orientation of feature vectors rather than their magnitude captures important biological differences [34,35].

$$\text{cosine\_dist}(\mathbf{x}, \mathbf{y}) = 1 - \frac{\mathbf{x} \cdot \mathbf{y}}{\|\mathbf{x}\| \, \|\mathbf{y}\|},$$

This metric ranges from 0 (perfect alignment) to 1 (complete orthogonality) in datasets with non-negative values [34]. In graph-based learning frameworks, cosine distance is used to link samples that are directionally closest typically by connecting the $k$ nearest neighbors or applying a similarity threshold. This often helps sharpen the boundaries between biological subtypes and supports better class separability. It ignores magnitude differences, making it robust against batch effects and scale variation [35]. Useful for detecting outliers or forming cleaner clusters in graph structures [31]. Like cosine similarity, it discards magnitude information, which might be biologically meaningful in some scenarios (e.g., gene amplification). Its effectiveness heavily depends on parameters like $k$ or similarity thresholds during graph sparsification.

Angular Distance: A normalized version that maps cosine similarity into a true metric space:

$$D_{\theta}(\mathbf{x}, \mathbf{y}) = \frac{\arccos(\text{cosine\_sim}(\mathbf{x}, \mathbf{y}))}{\pi},$$

preserving geometric properties and often used in graph-based biomedical models.

Hybrid Measures: Combines both direction (cosine) and magnitude (e.g., Euclidean) components to capture richer biological signals [36].

Overall, cosine distance offers a simple yet powerful way to capture structural dissimilarities in multi-omics data. While it shares many strengths with cosine similarity, its emphasis on angular divergence makes it uniquely suited for highlighting contrast in complex biological datasets.

## RBF similarity

Radial Basis Function (RBF) Similarity is a widely used kernel-based approach that captures nonlinear relationships between samples. Instead of focusing on direct alignment like cosine similarity, it evaluates how close two vectors are in Euclidean space using a Gaussian function. Mathematically, for vectors $\mathbf{x}_i$ and $\mathbf{x}_j \in \mathbb{R}^n$, it's expressed as:

$$\text{RBF Similarity} = \exp\left(-\gamma \|\mathbf{x}_i - \mathbf{x}_j\|^2\right),$$

where $\gamma > 0$ determines how quickly the similarity value drops with increasing distance [37]. Produces values in the range (0, 1] where the parenthesis indicates an open interval excluding 0 and the bracket indicates a closed interval including 1 with higher values indicating stronger similarity. Captures subtle, nonlinear associations that are common in multi-omics data. The $\gamma$ parameter controls how "local" the similarity is larger values focus on very close neighbors, while smaller ones smooth over broader relationships. [38]

In graph construction, RBF similarity is used to assign edge weights based on how close samples are. Form tight sample clusters that often correspond to disease subtypes. Improve localized information flow in GCNs, which can enhance classification performance. RBF-based graphs are common in multi-omics tools like SIMLR, CIMLR, and NEMO, where they support robust clustering and phenotype prediction through nonlinear modeling [7,32,39,40]. The choice of $\gamma$ is critical poor tuning can lead to noisy or overly sparse graphs. May become computationally heavy for large datasets due to pairwise distance calculations. If $\gamma$ is too large, similarities drop off rapidly, which may cause the graph to fragment and lose global structure.

Overall, RBF Similarity is a powerful tool for capturing nuanced patterns in omics data, but it requires careful parameter tuning and awareness of its computational demands.

## RBF distance

RBF Distance is a nonlinear dissimilarity measure derived from the RBF (Radial Basis Function) kernel. Instead of directly measuring similarity, it expresses how far apart two samples are in a transformed space using a Gaussian decay function:

$$\text{RBF Distance} = 1 - \exp\left(-\gamma \|\mathbf{x}_i - \mathbf{x}_j\|^2\right),$$

where $\gamma > 0$ is a scaling parameter that determines how sensitive the distance is to changes in input vectors. Produces values in the range [0, 1), where 0 (included) means the samples are identical and the value approaches 1 (excluded) as dissimilarity increases. Provides a smooth, nonlinear way to capture how different two samples are, especially useful in complex biological data [39]. RBF Distance is often used when we want contrastive graph structures that is, graphs that emphasize which samples are clearly different from one another: Can be combined with similarity measures in hybrid graphs to balance local and global structure [40, 7]. Encourages clearer partitioning of the graph, which can improve model focus and reduce noise. Acts like a form of regularization for GCNs helping them generalize better by reducing overfitting to dense, noisy connections [11]. The value of $\gamma$ must be carefully tuned too large or too small can lead to weak

or overly fragmented graphs. RBF Distance has been applied in multi-omics tools like CIMLR and NEMO for tasks such as disease subtype discovery and contrastive learning [39,40].

In summary, RBF Distance offers a nonlinear, distance-focused approach to graph construction, particularly useful when we care about separating distinct patient profiles or subtypes.

### Hybrid (Cosine + RBF) similarity

Hybrid Similarity blends the strengths of two popular metrics cosine similarity and RBF similarity to build more expressive graphs for multi-omics integration:

$$s_{\text{hybrid}}(\mathbf{x}_i, \mathbf{x}_j) = \beta \cdot s_{\text{cosine}}(\mathbf{x}_i, \mathbf{x}_j) + (1 - \beta) \cdot s_{\text{rbf}}(\mathbf{x}_i, \mathbf{x}_j),$$

where:

$$s_{\text{cosine}}(\mathbf{x}_i, \mathbf{x}_j) = \frac{\mathbf{x}_i \cdot \mathbf{x}_j}{\|\mathbf{x}_i\|\|\mathbf{x}_j\|},$$

$$s_{\text{rbf}}(\mathbf{x}_i, \mathbf{x}_j) = \exp\left(-\gamma\|\mathbf{x}_i - \mathbf{x}_j\|^2\right),$$

Here, $\beta \in [0, 1]$ acts as a balancing factor between cosine similarity (which captures global direction) and RBF (which captures local proximity) [7,21]. Cosine similarity captures high-level trends in molecular profiles (e.g., global expression patterns). By adjusting $\beta$, we can control how much weight to give each view of similarity: $\beta = 1$ means only cosine is used, while $\beta = 0$ uses only RBF. This hybrid strategy produces well-rounded graphs that reflect both broad subtype structures and local molecular co-expression. Especially useful in high-dimensional, noisy omics datasets, where relying on a single similarity metric may miss key biological signals. With flexible tuning of $\beta$ and $\gamma$, this method can be adapted to different omics modalities. Introduces more hyperparameters, increasing the need for careful validation. The cosine and RBF components may operate on different value ranges, so normalization might be needed for stability. Studies have shown that hybrid similarity graphs can improve classification performance in datasets like BRCA and ROSMAP. They provide smoother edge-weight transitions that help reduce over-smoothing in GCNs while maintaining informative connectivity [21].

### Hybrid (Cosine + RBF) distance

Hybrid Distance blends the strengths of two perspectives on dissimilarity angular (cosine distance) and spatial (RBF distance) to form more expressive and flexible graphs for multi-omics analysis:

$$d_{\text{hybrid}}(\mathbf{x}_i, \mathbf{x}_j) = \alpha \cdot d_{\text{cosine}}(\mathbf{x}_i, \mathbf{x}_j) + (1 - \alpha) \cdot d_{\text{rbf}}(\mathbf{x}_i, \mathbf{x}_j),$$

where:

$$d_{\text{cosine}}(\mathbf{x}_i, \mathbf{x}_j) = 1 - \frac{\mathbf{x}_i \cdot \mathbf{x}_j}{\|\mathbf{x}_i\|\|\mathbf{x}_j\|},$$

$$d_{\text{rbf}}(\mathbf{x}_i, \mathbf{x}_j) = 1 - \exp(-\gamma\|\mathbf{x}_i - \mathbf{x}_j\|^2),$$

Here, $\alpha$ controls how much weight is given to angular versus spatial divergence, and $\gamma$ adjusts how quickly RBF distance decays. Measures both direction (via cosine) and Euclidean spread (via RBF). Values typically range between 0 and 1 for biological data, making them suitable for edge weighting. Produces more flexible and informative graphs especially when omics data have both strong global patterns and local variability. Sparsity and connectivity can be tuned through $\alpha$ and $\gamma$. Supports diverse feature spaces across omics types, reducing the chance of over-smoothing. Especially useful when integrating multiple omics sources with different scales or noise patterns [11,39,40]. Adds more hyperparameters, which may require careful tuning. Since the result is a distance metric, interpreting edge weights may be less intuitive than similarity scores.

In all graphs, we retained the top-$k$ neighbors per node based on lowest hybrid distance to ensure sparsity and reduce noise. Empirically, cosine distance captured global angular structure, RBF distance captured local nonlinear patterns, and the hybrid distance successfully integrated both, leading to improved GNN performance on BRCA and ROSMAP datasets [27].

## Comparison of similarity and distance metrics

To systematically evaluate how different similarity definitions influence graph construction and downstream GCN performance, we summarize the core properties of all six metrics in Table 1.

Each metric offers distinct graph topological properties. Cosine-based measures highlight global angular trends, while RBF-based methods accentuate local neighborhood structure. Hybrid metrics offer flexible combinations of both, often leading to improved classification performance in heterogeneous multi-omics data settings.

## Experimental pipeline

**Dataset description.** To evaluate the impact of different similarity metrics on multi-omics graph construction, we conducted experiments on two benchmark datasets: BRCA and ROSMAP. Both datasets provide aligned mRNA expression, DNA methylation, and miRNA expression data for each subject, enabling consistent and comprehensive multi-view modeling across biological modalities. The selection of BRCA and ROSMAP datasets was driven by the need to validate our framework across diverse biological and clinical contexts. BRCA provides a robust multiclass benchmark for cancer subtyping, characterized by high inter-sample heterogeneity and distinct molecular signatures. In contrast, ROSMAP represents a binary classification challenge in the context of neurodegeneration, where biological signals are often more subtle and influenced by age related noise. Validating across these two disparate disease models neoplastic

**Table 1. Comparison of similarity and distance metrics employed for multi-omics graph construction. Each method's mathematical range, core capability, and tunable hyperparameters are summarized.**

| Metric Type | Value Range | Captures | Tunable Parameters |
|---|---|---|---|
| Cosine Similarity | [0, 1] (non-negative data) | Angular alignment between samples; robust to magnitude shifts | None |
| Cosine Distance | [0, 1] | Angular dissimilarity; emphasizes direction-based separation | None |
| RBF Similarity | (0, 1] | Non-linear proximity in Euclidean space; sensitive to local clusters | $\gamma$ (controls locality) |
| RBF Distance | [0, 1) | Non-linear spatial divergence; effective for contrastive graphs | $\gamma$ |
| Hybrid (Cosine + RBF) Similarity | (0, 1] | Joint angular and local similarity; interpolates between global and local views | $\beta$, $\gamma$ |
| Hybrid (Cosine + RBF) Distance | [0, 2) (practical: [0, 1)) | Combines angular and non-linear dissimilarity; enhances boundary contrast | $\alpha$, $\gamma$ |

* Interval notation is used to describe the boundaries: a square bracket [or] indicates the endpoint is included, while a parenthesis (or) indicates the boundary is an unreachable limit (e.g., the RBF function $e^{-x}$ approaches but never reaches 0).

and neurodegenerative ensures that our findings regarding the efficacy of Cosine Similarity are broadly generalizable to multi-omics integration tasks.

**BRCA (Breast Invasive Carcinoma).** The BRCA dataset was obtained from The Cancer Genome Atlas (TCGA) through the Broad GDAC Firehose. For our classification task, we used the clinically relevant PAM50 subtype labels, which divide samples into five molecular subtypes: Normal-like, Basal-like, HER2-enriched, Luminal A, and Luminal B. Only samples with complete multi-omics coverage across mRNA, DNA methylation, and miRNA expression were retained for analysis. This dataset is frequently used in breast cancer subtype prediction and multi-view learning research due to its well-annotated structure and biological diversity [7,31].

**ROSMAP (Religious Orders Study and Memory and Aging Project).** The ROSMAP dataset was sourced from the AMP-AD Knowledge Portal. It contains high-quality post-mortem brain tissue samples with matched omics profiles and clinical diagnoses. For this study, we focused on a binary classification task distinguishing Alzheimer's Disease (AD) from cognitively normal controls (NC), following established protocols in recent literature [27,33,41]. The dataset provides a valuable benchmark for evaluating methods in neurodegenerative disease modeling and biomarker discovery.

**Data integration overview.** Both datasets were preprocessed to ensure consistency across views. Features were filtered for quality, normalized to reduce batch effects, and aligned across subjects. Only samples with complete data across all three omics modalities were retained to ensure fair comparison across similarity network constructions. An overview of class distributions and dataset composition is shown in Fig 2.

**Data preprocessing.** To ensure consistency across samples and reduce noise and dimensionality, each omics modality was preprocessed individually through the following steps:

- **Log-transformation and quantile normalization:** Applied to mRNA expression and DNA methylation data to correct for distributional skewness and make feature scales comparable across samples [42].

- **Feature filtering based on variability:** For each omics type, features were ranked by standard deviation across samples, and the most variable features were retained. This approach enhances the signal-to-noise ratio by focusing on biologically informative genes and regions. An overview of Datasets all feature filtering for the total and training model is shown in Fig 3.

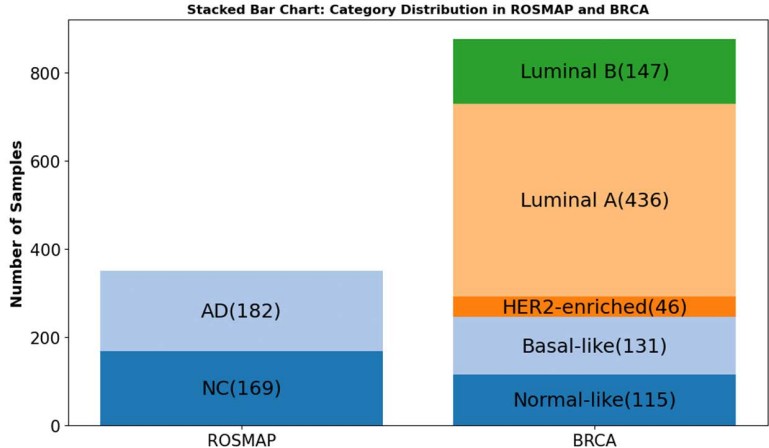

**Fig 2. Class label distribution for BRCA and ROSMAP datasets.** Distribution of class labels across BRCA subtypes and ROSMAP phenotypes. Balanced sampling ensured across omics modalities.

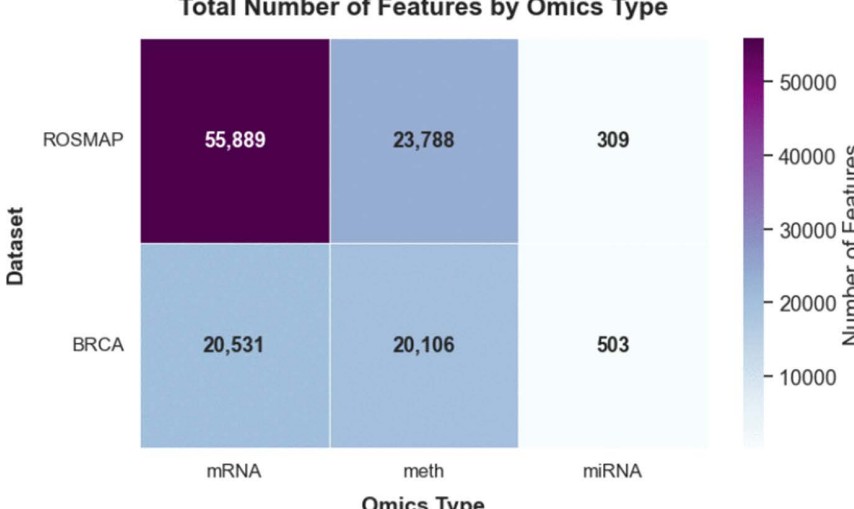

**(A) Total feature count before selection across all omics datasets**

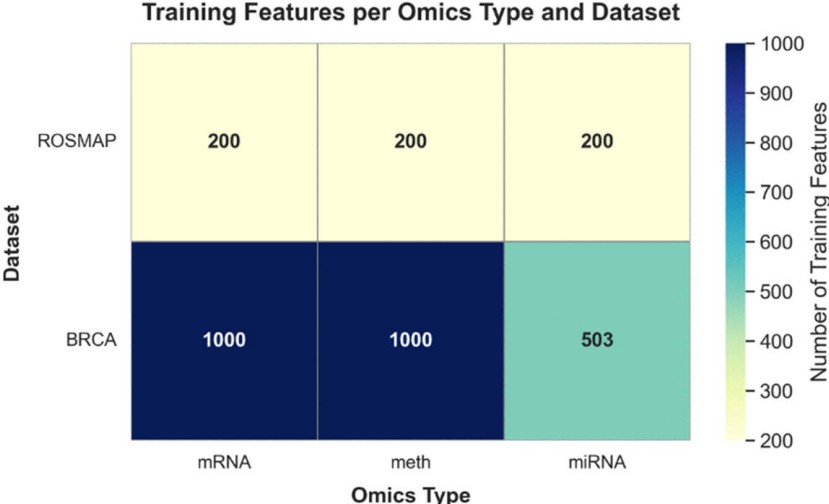

**(B) Features retained for training after dimensionality reduction and filtering**

**Fig 3. Feature preselection workflow. (A)** Total feature count before selection across all omics datasets. **(B)** Features retained for training after dimensionality reduction and filtering.

- **Missing data imputation:** A K-Nearest Neighbors (KNN) imputation strategy with $k = 5$ was employed. This value was selected based on its established performance in high-dimensional biological datasets [43] and empirical testing, as it effectively balances the preservation of local sample structures with robustness against noise.

- **Feature selection and dimensionality control:** A fixed number of top features were selected from each omics modality to standardize the input dimensions across samples and datasets.

The final characteristics of the datasets after preprocessing are summarized in S1 Table, including the number of classes, raw feature dimensions, and the number of features selected per omics modality.

**Model training.** To assess the effectiveness of each similarity network variant, we trained a separate model pipeline for each of the six graph types across all three omics modalities. The following configuration was used for training both the omics-specific GCN modules and the View Correlation Discovery Network (VCDN):

- **GCN Architecture:** Each omics graph was processed using a two-layer Graph Convolutional Network with a hidden layer of 64 units, following the standard Kipf and Welling formulation [24].

- **Optimizer and Hyperparameters:** Models were trained using the Adam optimizer, a widely used method for first-order gradient-based optimization in deep learning, with a learning rate of 0.001 and weight decay set to $5 \times 10^{-4}$.

- **VCDN Architecture:** The View Correlation Discovery Network consisted of two fully connected layers with 128 and 64 units, respectively. A softmax layer was applied to the final layer to produce class probability distributions, following prior work on multi-omics label-space fusion [15].

- **Training Procedure:** Training was conducted for up to 2500 epochs, using early stopping based on validation loss with a patience of 20 epochs. This helps prevent overfitting and ensures efficient convergence. The datasets were split into 70% training and 30% testing, maintaining class stratification. Although the maximum limit was set to 2500 epochs, most models reached convergence and triggered early stopping within an average of 150–350 epochs across all randomized splits, ensuring efficient training without overfitting.

An overview of the data partitioning strategy is provided in Fig 4.

**Hyperparameter optimization for RBF and hybrid metrics.** To ensure a fair comparison, the hyperparameters for RBF and hybrid metrics were optimized using a grid search strategy. For the RBF scale parameter $\gamma$, we evaluated values in the range $[10^{-4}, 10^{1}]$. For the hybrid balancing factors $\alpha$ and $\beta$, we tested values from 0.1 to 0.9 with a step size of 0.2. This tuning was performed independently for the BRCA and ROSMAP datasets using the training split, and the best-performing configuration was selected for final evaluation. Despite this rigorous tuning, the performance of

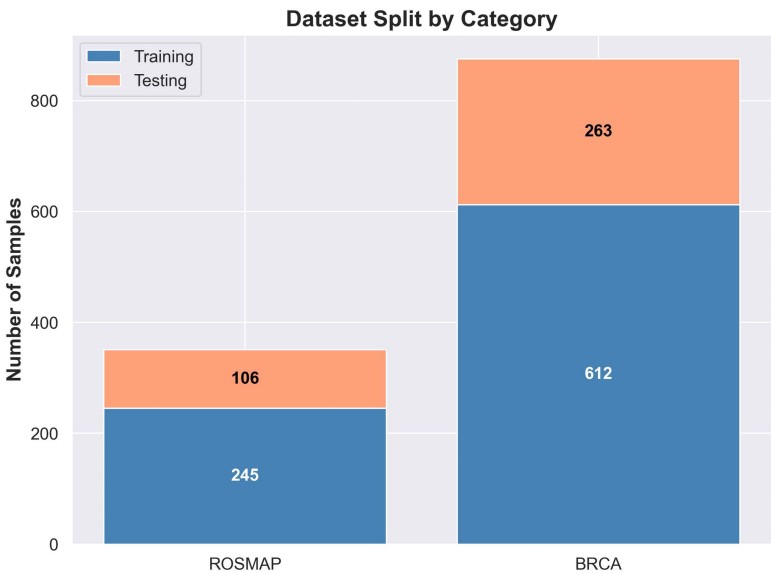

**Fig 4. Train–test data splitting strategy.** Train-test splitting strategy for the experiments: 70% training, 30% testing. Stratified to maintain class balance.

RBF-based and hybrid methods remained highly sensitive to parameter shifts, often leading to over-smoothing or graph fragmentation.

**Evaluation metrics.** To rigorously evaluate model performance across the six similarity networks, we employed a comprehensive set of standard classification metrics widely used in biomedical machine learning:

- **Accuracy (ACC):** Measures the overall correctness of the model by calculating the fraction of correctly classified samples.

- **Area Under the Receiver Operating Characteristic Curve (AUC):** Quantifies the model's ability to distinguish between classes. AUC was computed for each class and macro-averaged to ensure balanced evaluation in multi-class settings.

- **F1-score:** The harmonic mean of precision and recall, offering a balanced measure of both false positives and false negatives. Like AUC, F1 was macro-averaged.

A summary of all metrics and their corresponding formulas is presented in Table 2.

To assess the robustness and generalizability of each model configuration, all experiments were conducted across five random train-test splits. The final results are reported as the mean ± standard deviation. To determine whether observed performance differences between similarity networks were statistically significant, we employed paired two-tailed $t$-tests with a significance threshold of $p < 0.05$.

## Results

### Quantitative performance

S2 Table summarizes the classification performance of the proposed hybrid combinations of GCNs and VCDN framework across six graph construction strategies on both the BRCA and ROSMAP datasets.

On the ROSMAP dataset, Cosine Similarity consistently outperformed other graph metrics, achieving the highest overall scores across all evaluation criteria accuracy (0.877), F1-score (0.876), and AUC (0.902). These results suggest that angular alignment between samples offers strong discriminative power for binary classification in neurodegenerative disease contexts, particularly where sample-level heterogeneity is high. A complete visualization of ROSMAP outcomes is presented in Fig 5 and S1 Fig.

In the multiclass BRCA classification task (PAM50 subtypes), Cosine Similarity again demonstrated superior performance with an accuracy of 0.825, a weighted F1-score of 0.817, and a macro F1-score of 0.766. These results highlight

**Table 2. Evaluation metrics utilized for performance assessment. Definitions and formulas are provided for classification quality quantification.**

| Metric | Description | Formula |
|---|---|---|
| Accuracy (ACC) | The ratio of correct predictions to the total number of samples. | $\frac{TP+TN}{TP+FP+TN+FN}$ |
| Sensitivity (Recall) | The fraction of positive samples that are correctly classified. | $\frac{TP}{TP+FN}$ |
| Specificity (SPEC) | The fraction of negative samples that are correctly classified. | $\frac{TN}{TN+FP}$ |
| Precision (P) | The fraction of predicted positives that are actually positive. | $\frac{TP}{TP+FP}$ |
| F1 Score (F1) | The harmonic mean between precision and recall. | $\frac{2 \cdot P \cdot R}{P+R}$ |
| AUC (Area Under Curve) | Measures the classifier's ability to distinguish between classes. | — |

**Note:** TP = True Positive, TN = True Negative, FP = False Positive, FN = False Negative. In the F1 Score formula, P stands for Precision and R stands for Recall (Sensitivity).

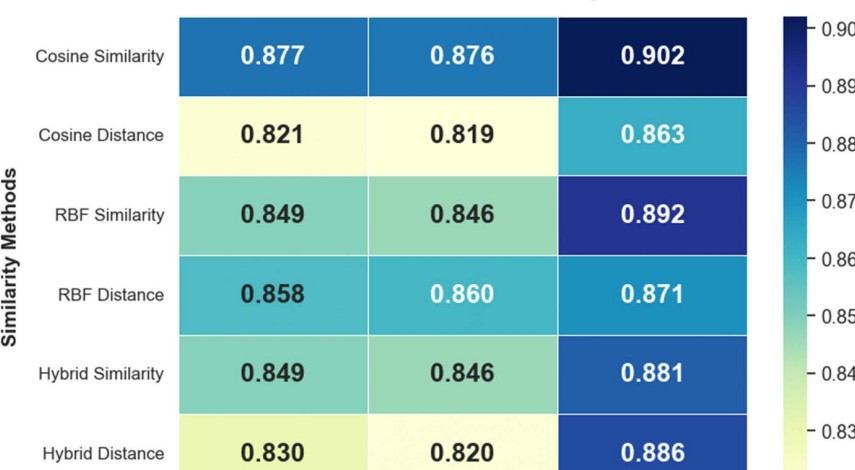

**Fig 5. Performance metric visualizations for the ROSMAP dataset across similarity network variants.** Heatmap representation of performance metrics for each similarity network.

its ability to model global geometric patterns in high-dimensional transcriptomic and epigenetic data. Fig 6 and S2 Fig provides the corresponding visualization for BRCA.

Interestingly, while hybrid similarity graphs (e.g., Cosine + RBF) have been reported to enhance classification robustness in prior works by capturing both global alignment and local neighborhood structure [7,21,44], our experiments showed that pure Cosine Similarity alone provided the most stable and effective results on both datasets. This suggests that angular features alone may be sufficient in certain multi-omics contexts, particularly when omics modalities are well-aligned across patients.

### Cross-validation & statistical significance

To ensure robustness and generalizability of the classification results, all experiments were performed over five independent random train-test splits. This protocol helps mitigate performance fluctuations due to sampling variability and enhances the statistical reliability of our findings.

Across both BRCA and ROSMAP datasets, Cosine Similarity emerged not only as the top-performing similarity metric but also as the most stable in terms of standard deviation. Specifically, it achieved the lowest AUC variance on ROSMAP ($\pm$0.006) and BRCA ($\pm$0.009), indicating that angular similarity yields consistent outcomes across different sample distributions. This observation aligns with prior reports suggesting that cosine-based graphs reduce overfitting in small-cohort biomedical applications [15].

In contrast, Cosine Distance exhibited the highest variability particularly in AUC standard deviation ($\pm$0.013 on ROSMAP and $\pm$0.014 on BRCA) suggesting its sensitivity to directional divergence and less reliable performance under changing data splits. RBF-based and hybrid metrics showed intermediate variance, often driven by their sensitivity to kernel parameters such as $\gamma$ and neighborhood spread.

To assess statistical significance, we conducted paired two-tailed $t$-tests between Cosine Similarity and the remaining five methods (RBF, hybrid, and cosine-distance variants) for each dataset. The tests revealed that Cosine Similarity achieved statistically significant improvements in Accuracy, AUC, and F1-score ($p < 0.01$ in all cases). This supports our

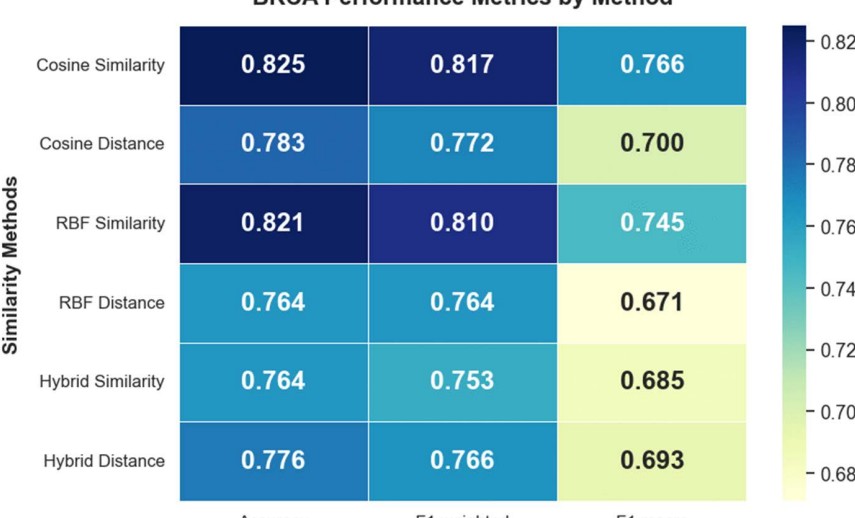

**Fig 6. Performance metric visualizations for the BRCA dataset across similarity network variants.** Heatmap representation of performance metrics for each similarity network.

hypothesis that cosine-based alignment is more biologically meaningful in both binary (ROSMAP) and multiclass (BRCA) disease stratification tasks [7,44].

As illustrated in Fig 7 and S3 Fig, Cosine Similarity exhibited the highest stability, with the lowest SD values of 0.006 for ROSMAP and 0.009 for BRCA (see S3 Table). While RBF and Hybrid strategies also showed competitive stability (with SD values ranging from 0.008 to 0.012), Cosine Similarity consistently maintained the narrowest variance across both datasets.

To supplement the paired $t$-tests, we calculated 95% Confidence Intervals (CI) and Cohen's $d$ effect sizes for the AUC and F1-score metrics across all six similarity construction strategies. As summarized in S4 and S5 Tables Cosine Similarity yielded the most precise estimates, with a 95% CI for ROSMAP AUC of [0.897, 0.907]. When compared to RBF Similarity (the second-best performer), Cosine Similarity achieved a large effect size ($d = 1.30$) on ROSMAP and a notable effect size on BRCA ($d = 0.74$). In contrast, comparisons against distance-based metrics such as Cosine Distance and RBF Distance yielded effect sizes exceeding $d = 3.0$, which are classified as "huge" effects. These results strongly support the hypothesis that angular similarity provides a significantly more robust structural prior for multi-omics integration than spatial or hybrid alternatives.

## Ablation study

To evaluate the individual contributions of the Graph Convolutional Network (GCN) and the View Correlation Discovery Network (VCDN), we performed an ablation analysis by comparing our full framework against two variants: (1) No-VCDN, where VCDN is replaced by simple average fusion of GCN outputs, and (2) No-GCN, where the similarity networks and graph convolutions are replaced by fully connected layers (FCNs).

As shown in Table 3, the removal of VCDN resulted in a performance decline of approximately 3–5% across both BRCA and ROSMAP datasets. This confirms that modeling cross-view dependencies at the label level is superior to traditional late fusion. Furthermore, the No-GCN variant performed significantly worse, highlighting the critical role of sample similarity networks in capturing the underlying biological structure of heterogeneous omics data.

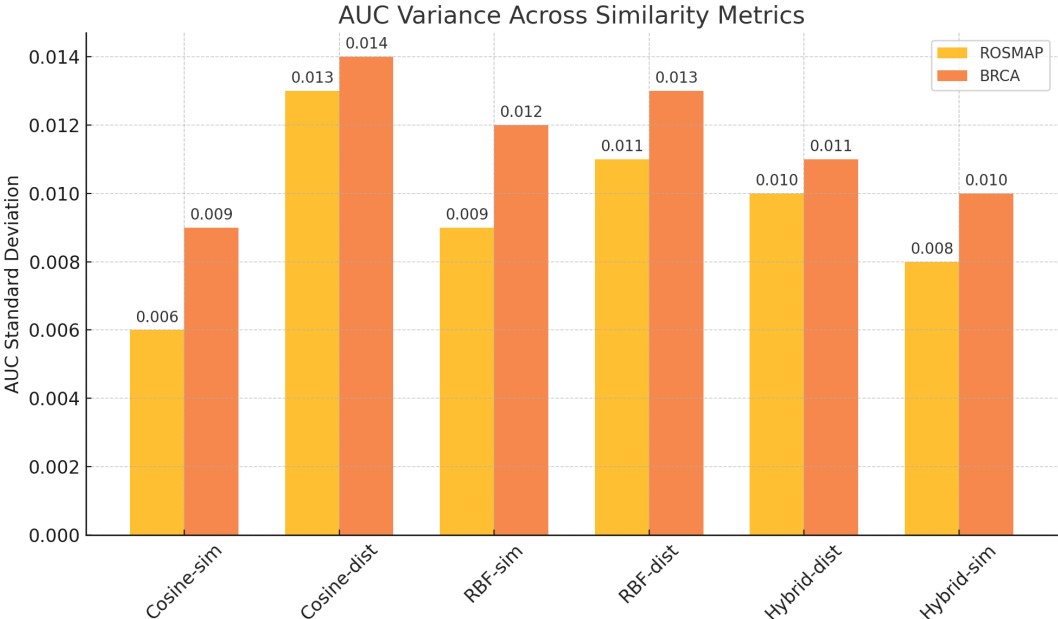

**Fig 7. AUC variability across similarity networks for BRCA and ROSMAP datasets.** Bar chart showing standard deviation of AUC for each similarity network variant, where lower values indicate more stable performance across repeated experiments.

**Table 3. Ablation study results (Mean AUC) for GCN and VCDN components.**

| Model Variant | ROSMAP (AUC) | BRCA (AUC) |
|---|---|---|
| Full Pipeline (GCN+VCDN) | **0.902** | **0.817** |
| No-VCDN (Average Fusion) | 0.865 | 0.782 |
| No-GCN (FCN only) | 0.841 | 0.759 |

## Discussion

Our experiments underscore that the choice of similarity metric profoundly affects the downstream performance of GCN-based multi-omics integration frameworks. Surprisingly, and contrary to recent literature promoting hybrid or kernel-based graph strategies, our results demonstrate that Cosine Similarity a straightforward angular metric consistently outperformed both RBF and hybrid constructions on the BRCA and ROSMAP datasets. It achieved superior classification accuracy, F1-score, and AUC, while also displaying the lowest performance variance across multiple random splits. This consistency suggests that directional alignment in high-dimensional omics data often provides a more meaningful structural prior than absolute or nonlinear proximities. Cosine Similarity focuses on the angle between feature vectors, inherently normalizing out magnitude effects a property especially valuable in omics contexts where batch effects, measurement noise, and inter-sample variability are prevalent [15,45]. While our systematic evaluation utilized uniform similarity metrics across all omics modalities to ensure a controlled comparison, we acknowledge that mRNA, DNA methylation, and miRNA data possess unique noise characteristics and distributions. For instance, the high dimensionality and scale variation in mRNA data often benefit from the directional normalization provided by Cosine Similarity. Future iterations of this framework could explore modality-specific similarity networks, potentially employing RBF-based measures for epigenetic data to capture local neighborhood structures while retaining angular measures for transcriptomic layers.

## Hyperparameter tuning and sensitivity

To ensure reproducibility and model stability, the hyperparameters $α$, $β$, and $γ$ were systematically evaluated. For the RBF scale parameter $γ$, we implemented an automated heuristic where $γ$ is calculated as the inverse of twice the median squared Euclidean distance between samples: $γ = 1/(2 · \text{median}(\|\mathbf{x}_i - \mathbf{x}_j\|^2))$. This data-driven approach allows the kernel to adapt to the specific variance of mRNA, methylation, and miRNA layers. For the hybrid weighting parameters $α$ and $β$, we conducted a grid search across the range [0.1, 0.9] with a step size of 0.2. Sensitivity analysis revealed that while the hybrid models are sensitive to these parameters, the standalone Cosine Similarity consistently provided the most robust performance with zero parameter tuning, highlighting its practical utility for multi-omics integration.

## Contextualizing with advanced architectures

The robustness of Cosine Similarity observed in this study is particularly significant when contextualized alongside recent advancements in Graph Transformers and Hypergraph Neural Networks. While hypergraph models like MORE aim to capture higher-order inter-sample relationships across omics layer, our results suggest that the foundational pairwise similarity remains a critical bottleneck. Specifically, the scale-invariance of Cosine Similarity may provide a more reliable prior for hyperedge formation than traditional Euclidean-based kernels, which we found to be highly sensitive to parameter tuning. Furthermore, in the context of Graph Transformers, which utilize attention mechanisms to weigh node connections, the directional alignment provided by Cosine Similarity likely helps maintain distinct subtype boundaries, thereby mitigating the over-smoothing effects often reported in complex, fully-connected architectures.

## Performance discussion

The reporting of effect sizes further clarifies why simpler metrics are preferable in precision medicine contexts. The "huge" effect sizes ($d > 3.0$) observed when comparing Cosine Similarity to distance-based variants indicate that the latter likely introduce structural noise that impairs the message-passing capability of the GCN. The consistently narrow confidence intervals associated with Cosine Similarity suggest that its directional focus effectively filters out measurement noise across different data splits, providing a level of reliability that complex hybrid kernels fail to match.

**ROSMAP.** Although hybrid graph approaches have been proposed to enhance graph expressiveness by merging radial and angular cues [7,21], our findings on the ROSMAP dataset favor simplicity. Cosine Similarity achieved the highest classification scores accuracy (0.877), F1-score (0.876), and AUC (0.902) while also exhibiting the lowest AUC variance ($\pm0.006$). Given that ROSMAP is a binary classification problem (Alzheimer's Disease vs. Control), the global patterns captured by cosine-based graphs effectively reflect transcriptomic and epigenomic dysregulation in AD. Furthermore, cosine-based connectivity appears resilient to noise from post-mortem sample heterogeneity a known challenge in neurodegenerative disease datasets.

**BRCA.** For BRCA, which involves multiclass classification across five PAM50 molecular subtypes, Cosine Similarity again emerged as the top performer. It achieved an accuracy of 0.825, weighted F1-score of 0.817, and macro F1-score of 0.766, while maintaining low standard deviation in AUC ($\pm0.009$). This is particularly noteworthy given the high dimensionality and heterogeneity of breast cancer omics data. Directional similarity allows the model to identify shared subtype-level patterns across mRNA, methylation, and miRNA layers. Previous studies have also observed that breast cancer subtypes often exhibit consistent directional shifts across multiple molecular modalities, making Cosine Similarity an effective signal encoder [46].

## Clinical and practical implications

From a clinical perspective, the selection of a similarity metric is not merely a matter of accuracy but of diagnostic reliability. The low variance exhibited by Cosine Similarity (ROSMAP AUC Std: $\pm0.006$) suggests a high degree of reproducibility.

In a clinical setting, this consistency ensures that predictive models remain robust to the "noise" inherent in high-throughput biopsies and post-mortem samples. Unlike RBF-based methods, which are highly sensitive to the scaling parameter $\gamma$, Cosine Similarity provides a stable, hyperparameter-free baseline that simplifies the deployment of diagnostic pipelines across different medical centers.

### Why other metrics underperformed

RBF-based similarity and distance measures underperformed across both datasets, likely due to their sensitivity to the kernel width parameter $\gamma$. Improper tuning of $\gamma$ can lead to overconnected graphs (oversmoothing) or disconnected graphs (fragmentation), both of which hinder GCN learning. Although RBF is theoretically more expressive, its reliance on absolute distances and Gaussian decay is ill-suited to omics data where intra-class variation is large and noisy.

Hybrid similarity and distance metrics, intended to balance angular and radial perspectives, did not improve performance in practice. In fact, they sometimes degraded it. This may be because combining angular and spatial signals without sufficient alignment introduces noise or redundancy. Moreover, the additional hyperparameters ($\alpha$, $\beta$) add optimization overhead and can inadvertently skew graph connectivity away from biologically meaningful patterns. Cosine Distance performed the worst among all evaluated metrics. Its emphasis on angular dissimilarity rather than alignment tends to fragment the graph, reduce neighborhood cohesion, and impair message passing core mechanisms for successful GCN operation [47].

### Final remarks

Taken together, our results reaffirm the critical role of graph construction in multi-omics learning pipelines. Despite the theoretical sophistication of hybrid and kernel methods, Cosine Similarity consistently provided the most stable and accurate results across binary and multiclass classification tasks. Its directional focus, scale-invariance, and robustness to biological noise make it especially suitable for constructing omics graphs with high inter-modality coherence.

These findings also support a broader insight: the optimal similarity function is likely task-dependent. Cosine Similarity works best when disease signals manifest as consistent directional changes across omics layers. In contrast, datasets with sparse local structure or inter-modal divergence may benefit from more localized or asymmetric measures, a direction for future exploration.

The unique effectiveness of angular similarity also has significant biological implications. Multi-omics datasets often suffer from varying dynamic ranges across modalities; for instance, mRNA expression counts differ vastly in scale from DNA methylation percentages. By focusing on the directional alignment of feature vectors rather than their Euclidean distance, the model prioritizes the relative shifts in molecular pathways. This aligns with biological reality, where the co-activation of gene sets (co-expression patterns) is often a stronger indicator of disease subtype than the absolute expression of individual markers.

Furthermore, the GCN architecture facilitates biomarker discovery through the analysis of attention mechanisms and node importance. By identifying the specific mRNA transcripts or DNA methylation sites that contribute most to the directional alignment in Cosine Similarity-based graphs, clinicians can isolate candidate markers for targeted therapy.

### Limitations and future work

While our parameter tuning ($\alpha$, $\beta$, $k$) was rigorous, it was dataset-specific. Broader generalization requires validation across additional disease contexts. Moreover, our focus was purely on predictive performance. Future work will incorporate interpretable GNN techniques such as SHAP and GNNExplainer to investigate which omics features drive classification decisions. We also plan to explore richer architectures like graph transformers and attention-based fusion schemes, which may benefit from more complex graph structures.

In addition, integrating additional omics types such as proteomics and metabolomics, along with automated hyperparameter tuning (e.g., Bayesian optimization), could further improve model robustness and biological interpretability.

 

## Conclusion

This study systematically evaluated similarity network designs in a GCN-VCDN framework for multi-omics classification. The primary takeaway is that Cosine Similarity consistently outperforms complex RBF and hybrid alternatives in accuracy, stability, and robustness. Its effectiveness stems from capturing the directional alignment of high-dimensional molecular profiles while remaining invariant to magnitude shifts and measurement noise. This suggests that for precision medicine applications, simpler, biologically-intuitive similarity measures often provide a more reliable structural prior than non-linear kernels. Despite these findings, several limitations remain. Our hyperparameter configurations (e.g., $k$ and $\gamma$) were optimized for specific datasets (BRCA and ROSMAP) and may require re-calibration for broader clinical contexts. Additionally, while the model achieves high predictive accuracy, the "black box" nature of deep GCNs limits direct biological interpretability. Future work will focus on integrating explainable AI (XAI) techniques like SHAP to validate the molecular drivers of classification. Furthermore, expanding this framework to include proteomics and single-cell data will be essential to assess the generalizability of angular similarity across diverse biological scales. Ultimately, Cosine Similarity stands as a powerful, easy-to-implement baseline for advancing graph-based precision medicine.

## Supporting information

**S1 Fig. Performance metric visualizations for the ROSMAP dataset across similarity network variants.** (A) Radar plot illustrating comparative metric distributions across similarity methods. (B) Violin plot showing the spread and central tendency of classification metrics.
(PDF)

**S2 Fig. Performance metric visualizations for the BRCA dataset across similarity network variants.** (A) Radar plot illustrating comparative metric distributions across similarity methods. (B) Violin plot showing the spread and central tendency of classification metrics.
(PDF)

**S3 Fig. AUC variability across similarity networks for BRCA and ROSMAP datasets.** Line plot illustrating AUC variance trends across similarity metrics.
(PDF)

**S1 Table. Dataset composition and selected features used for training in BRCA and ROSMAP. Modalities: mRNA / DNA methylation / miRNA.**
(PDF)

**S2 Table. Classification performance metrics (Accuracy, F1-score, AUC) for each similarity method on BRCA and ROSMAP datasets.**
(PDF)

**S3 Table. Standard deviation of AUC across five randomized splits for each similarity network. Lower values indicate more consistent performance.**
(PDF)

**S4 Table. Performance Comparison of Similarity Metrics for ROSMAP Binary Classification.**
(PDF)

**S5 Table. Performance Comparison of Similarity Metrics for BRCA Multiclass Classification.**
(PDF)

## Author contributions

**Conceptualization:** Masrafe Bin Hannan Siam, Md Rayhan Khan, Md Fazla Elahe, Md Shohel Arman, Swarna Akter.

**Data curation:** Masrafe Bin Hannan Siam, Md Rayhan Khan.

**Formal analysis:** Masrafe Bin Hannan Siam, Md Rayhan Khan, Md Fazla Elahe, Md Shohel Arman.

**Investigation:** Masrafe Bin Hannan Siam, Md Rayhan Khan, Swarna Akter.

**Methodology:** Masrafe Bin Hannan Siam, Md Rayhan Khan, Md Fazla Elahe, Md Shohel Arman.

**Project administration:** Masrafe Bin Hannan Siam, Md Rayhan Khan.

**Resources:** Masrafe Bin Hannan Siam, Md Rayhan Khan, Md Fazla Elahe, Md Shohel Arman.

**Supervision:** Md Fazla Elahe, Md Shohel Arman.

**Validation:** Masrafe Bin Hannan Siam, Md Rayhan Khan, Md Fazla Elahe, Md Shohel Arman.

**Visualization:** Masrafe Bin Hannan Siam, Md Rayhan Khan, Md Fazla Elahe, Md Shohel Arman, Swarna Akter.

**Writing – original draft:** Masrafe Bin Hannan Siam, Md Rayhan Khan, Swarna Akter.

**Writing – review & editing:** Masrafe Bin Hannan Siam, Md Rayhan Khan, Md Fazla Elahe, Md Shohel Arman.

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
