## [Decision Letter · Decision Letter 0]

26 Nov 2025

Dear Dr. ELAHE,

Thank you for submitting your manuscript to PLOS ONE. After careful consideration, we feel that it has merit but does not fully meet PLOS ONE’s publication criteria as it currently stands. Therefore, we invite you to submit a revised version of the manuscript that addresses the points raised during the review process.

We look forward to receiving your revised manuscript.

Kind regards,

Junhuang Jiang

Academic Editor

PLOS ONE

Journal Requirements:

2. We note that your Data Availability Statement is currently as follows: All relevant data are within the manuscript and Supporting Information files

Reviewers' comments:

Reviewer's Responses to Questions

**Comments to the Author**

1. Is the manuscript technically sound, and do the data support the conclusions?

Reviewer #1: Yes

Reviewer #2: Yes

Reviewer #3: Yes

Reviewer #4: Yes

2. Has the statistical analysis been performed appropriately and rigorously?

Reviewer #1: No

Reviewer #2: Yes

Reviewer #3: Yes

Reviewer #4: Yes

3. Have the authors made all data underlying the findings in their manuscript fully available?

Reviewer #1: Yes

Reviewer #2: Yes

Reviewer #3: Yes

Reviewer #4: Yes

4. Is the manuscript presented in an intelligible fashion and written in standard English?

Reviewer #1: Yes

Reviewer #2: Yes

Reviewer #3: Yes

Reviewer #4: Yes

Reviewer #1: • The manuscript is well-structured but requires deeper justification for selecting only BRCA and ROSMAP datasets to validate generalizability.

• Figures illustrating the experimental workflow and results could be improved with clearer legends and higher resolution for readability.

• The discussion section should explicitly compare findings with recent graph-transformer and hypergraph models to contextualize novelty.

• Parameter sensitivity analysis for α, β, and γ is missing and should be included to enhance reproducibility.

• The comparison with state-of-the-art models is limited; include some recent work such as Feature selection using game Shapley improved grey wolf optimizer for optimizing cancer classification, A hybrid feature gene selection approach by integrating variance filter, extremely randomized tree, and Cuckoo Search algorithm for cancer classification, AI-driven insights: a machine learning approach to lung cancer diagnosis, Efficient gene selection for breast cancer classification using Brownian Motion Search Algorithm and Support Vector Machine, Hybrid Feature Selection Techniques Utilizing Soft Computing Methods for Classifying Microarray Cancer Data, Improved GA based Clustering with a New Selection Method for Categorical Dental Data, A Novel Technique for Image Captioning Based on Hierarchical Clustering and Deep Learning, Study of Optimality Strategies for Two-Person Game Model Under Interval Uncertainty.

• The introduction could better emphasize the research gap and how this study advances beyond existing GCN-based multi-omics works.

• The statistical validation (t-tests) should be complemented by effect size or confidence interval reporting to strengthen the results.

• The authors may include an ablation study to show the individual contributions of GCN and VCDN components.

• The manuscript would benefit from highlighting biological interpretability and practical implications of the results in clinical contexts.

• The conclusion section should be condensed and focus more on key takeaways and limitations rather than re-summarizing results.

• English language and grammar are generally good but could be polished for conciseness and clarity, especially in the methodology section.

Reviewer #2: In this manuscript, Bin Hannan Siam and colleagues have independently evaluated the classification performance (accuracy) of six similarity network construction strategies, which are implemented in graph convolutional networks (GCNs), a graph-based deep learning approach, using two benchmark datasets, BRCA and ROSMAP. The authors conclude that Cosine Similarity, one of the six similarity network construction strategies mentioned above, outperforms the others by demonstrating the superior accuracy, F1 score, and AUC, with the lowest standard deviation across validation splits using GCNs constructed with multiple omics datasets, including mRNA expression, DNA methylation, and miRNA expression.

Overall, this manuscript is well written and well structured, providing profound information about the theoretical rationales and the analytical workflow. The finding and the conclusion are straightforward and sound. In this circumstance, only several minor comments from my perspective are for consideration.

Minor comments:

1. “(0,1]” is written at several places, including lines 271, 302, and Table 1. I cannot recall whether a hybrid of half parentheses and brackets brings a specific mathematical meaning. If not, please correct it.

2. Please provide the citation for the statement mentioned in line 274.

3. Line 434: Can the authors elaborate more about the choice of k equal to 5? Is it based on empirical observations, or have the authors tested other sizes of k?

4. Please provide a Table legend for Table 3, explaining the representation of TP (true positive?), TN, FP, FN, P, and R written in the formulas.

Reviewer #3: General Comments:

This manuscript presents a systematic evaluation of similarity-network construction strategies within a graph-based multi-omics classification framework. Overall, the study is clearly motivated and generally well written. The finding that the simple Cosine Similarity outperforms more complex RBF and hybrid metrics is significant and provides valuable, practical guidance for the field. But several issues regarding methodological clarity, redundancy in presentation, and justification of analytical choices should be addressed before the manuscript can be considered for publication.

Specific Comments:

1. Line 118: The manuscript states that “five similarity network strategies” were evaluated, whereas in other parts six strategies are described (Cosine similarity, Cosine distance, RBF similarity, RBF distance, Hybrid similarity, Hybrid distance).

2. The Methods section describing the similarity metrics is excessively detailed and interrupts the flow. I recommend keeping the essential definitions and moving extended mathematical explanations or background text to the Supplementary Material.

3. The authors should clarify how the number of selected features (e.g., 1000 / 1000 / 503 for BRCA; 200 / 200 / 200 for ROSMAP) was determined. Have different feature counts been tested? How does the number of selected features influence model stability or accuracy?

4. Several tables duplicate information already shown in figures: Table 2 duplicates Fig. 3, Table 4 duplicates Fig. 5 and Fig. 6, Table 5, Fig. 8A, and Fig. 8B show highly overlapping content. I suggest removing the redundant tables or figures to improve readability. Fig. 7 does not add additional information beyond Fig. 5 and Fig. 6. If the authors want to keep it, including cross-validation variance, standard deviation, or confidence intervals would make the figure more meaningful. Otherwise, I suggest removing it.

5. It would be valuable for the authors to comment on whether certain similarity metrics work better for specific omics types (mRNA vs. methylation vs. miRNA). Different omics layers have distinct distributions and noise characteristics. Testing modality-specific similarity networks could provide additional insight.

6. Because the hybrid and RBF methods depend heavily on γ and weighting parameters, the manuscript should describe the tuning procedures, the search space used, and whether the tuning strategy was applied consistently across all datasets.

7. Although early stopping is used, 2,500 epochs is relatively high for small omics datasets. The authors could provide the average stopping epoch and include validation-loss curves in the supplementary materials.

8. Since six networks * three omics * cross-view training were performed, the authors should provide runtime comparisons and discuss the computational resources required.

Reviewer #4: The manuscript presents a well-designed and interesting study.

However, after careful consideration, I feel that the main focus of the research does not fully align with the scope of PLOS ONE.

While the findings are valuable within its specific field, they do not sufficiently meet the journal’s criteria regarding broad scientific relevance.

This is purely a matter of scope rather than the quality or significance of the study itself. I believe the manuscript may be better suited for a more specialized journal that focuses directly on this type of research.

**Do you want your identity to be public for this peer review?** For information about this choice, including consent withdrawal, please see our Privacy Policy

Reviewer #1: No

Reviewer #2: **Yes:** Heng-Chang Chen

Reviewer #3: No

Reviewer #4: No

---

## [Author Response · Author response to Decision Letter 1]

12 Jan 2026

The authors thank the reviewer for the positive assessment of our study's quality and significance. We respectfully address the concern regarding the scope by highlighting the broader scientific relevance of our findings. While our application focuses on BRCA and ROSMAP datasets, the core discovery that simple angular similarity can be more robust and accurate than complex kernel-based or hybrid methods in high-dimensional graph-based learning addresses a fundamental challenge in Geometric Deep Learning. This insight is highly relevant to any scientific field dealing with high-dimensional, noisy, and heterogeneous data, such as social network analysis, recommendation systems, and general signal processing. Furthermore, PLOS ONE frequently publishes comparative methodological studies that provide 'practical guidance for the field'. By demonstrating a robust baseline that reduces computational overhead and improves interpretability, our work provides a foundational 'best practice' for the growing community of researchers applying Graph Convolutional Networks (GCNs) to diverse biological and non-biological domains. We believe this broad applicability aligns well with the journal’s mission to publish technically sound research with wide-reaching impact.

---

## [Decision Letter · Decision Letter 1]

30 Jan 2026

Dear Dr. ELAHE,

Thank you for submitting your manuscript to PLOS ONE. After careful consideration, we feel that it has merit but does not fully meet PLOS ONE’s publication criteria as it currently stands. Therefore, we invite you to submit a revised version of the manuscript that addresses the points raised during the review process.

We look forward to receiving your revised manuscript.

Kind regards,

Junhuang Jiang

Academic Editor

PLOS One

Journal Requirements:

Reviewers' comments:

Reviewer's Responses to Questions

**Comments to the Author**

Reviewer #1: All comments have been addressed

Reviewer #2: All comments have been addressed

Reviewer #3: (No Response)

2. Is the manuscript technically sound, and do the data support the conclusions?

Reviewer #1: Yes

Reviewer #2: Yes

Reviewer #3: Yes

3. Has the statistical analysis been performed appropriately and rigorously?

Reviewer #1: Yes

Reviewer #2: Yes

Reviewer #3: Yes

4. Have the authors made all data underlying the findings in their manuscript fully available?

Reviewer #1: Yes

Reviewer #2: Yes

Reviewer #3: Yes

5. Is the manuscript presented in an intelligible fashion and written in standard English?

Reviewer #1: Yes

Reviewer #2: Yes

Reviewer #3: Yes

Reviewer #1: None,the authors have adequately addressed all the comments raised in the previous round of review. I find the revised manuscript acceptable for publication.

Reviewer #2: The authors have carefully addressed all my comments. At this stage, I don't have any more comments, so I recommend this manuscript for publication.

Reviewer #3: The authors have carefully considered and adequately addressed my main concerns from the previous round. The manuscript has improved substantially in clarity, organization, and methodological justification. At this stage, I have no major scientific concerns. My remaining comments mainly relate to presentation and redundancy. There is still considerable overlap in the information presented across multiple figures and tables, which affects the readability and conciseness of the manuscript. By implementing some consolidations, the core message—that Cosine Similarity is the most robust and accurate metric —will be supported by a much cleaner and more professional presentation.

In particular, Figure 5 and Figure 6, both figures use three different visual formats (Radar plot, Violin plot, and Heatmap) to present the same set of performance metrics (Accuracy, F1, and AUC). Keep only one sub-figure (for example Figure 5C and 6C) per dataset is recommended.

Tables 3 and 4 vs. Figures 5 and 6: Tables 3 and 4 provide Mean AUC, SD, and Cohen’s d. If the Heatmaps (5C/6C) are kept, consider moving the "Cohen’s d" and "95% CI" columns to the text or a supplementary table to avoid repeating the "Mean AUC" and "SD" values cross three different locations (Table, Figure Heatmap, and Supporting Information).

Figure 7 presents the standard deviation of AUC using both a line plot (A) and a bar chart (B). I recommend keep only Figure 7B (Bar Chart). Ensure the numerical values from S3 Table (AUC standard deviation) are cited in the text so that the visual trend in Figure 7B is sufficient on its own.

**Do you want your identity to be public for this peer review?** For information about this choice, including consent withdrawal, please see our Privacy Policy

Reviewer #1: No

Reviewer #2: **Yes:** Heng-Chang Chen

Reviewer #3: No

---

## [Author Response · Author response to Decision Letter 2]

31 Jan 2026

The authors thank the Reviewers for their careful review of our revisions and for the recommendation for publication.

---

## [Decision Letter · Decision Letter 2]

24 Feb 2026

Effects of Similarity Networks in Graph-Based Multi-Omics Classification

PONE-D-25-45745R2

Dear Dr. ELAHE,

We’re pleased to inform you that your manuscript has been judged scientifically suitable for publication and will be formally accepted for publication once it meets all outstanding technical requirements.

Kind regards,

Tao Huang

Academic Editor

PLOS One

Additional Editor Comments (optional):

Reviewers' comments:

Reviewer's Responses to Questions

**Comments to the Author**

Reviewer #1: All comments have been addressed

Reviewer #2: All comments have been addressed

Reviewer #3: All comments have been addressed

2. Is the manuscript technically sound, and do the data support the conclusions?

Reviewer #1: Yes

Reviewer #2: Yes

Reviewer #3: Yes

3. Has the statistical analysis been performed appropriately and rigorously?

Reviewer #1: No

Reviewer #2: Yes

Reviewer #3: Yes

4. Have the authors made all data underlying the findings in their manuscript fully available?

Reviewer #1: Yes

Reviewer #2: Yes

Reviewer #3: Yes

5. Is the manuscript presented in an intelligible fashion and written in standard English?

Reviewer #1: Yes

Reviewer #2: (No Response)

Reviewer #3: Yes

Reviewer #1: None, The manuscript presents a systematic and well-designed evaluation of similarity network construction strategies for graph-based multi-omics disease classification. The comparative analysis across BRCA and ROSMAP datasets is comprehensive and statistically sound. A key and interesting finding is that simple Cosine Similarity consistently outperforms more complex kernel-based and hybrid approaches in accuracy, stability, and robustness. The study highlights the importance of interpretability and simplicity in high-dimensional biological data modeling. No concerns regarding research ethics, data usage, or dual publication are observed.

Reviewer #2: The authors have greatly improved the quality of this manuscript in terms of readability of the main text and figures. I do not have further comments and recommend publication at PLOS One.

Reviewer #3: All of my previous concerns have been addressed satisfactorily, and I have no additional comments at this time.

**Do you want your identity to be public for this peer review?** For information about this choice, including consent withdrawal, please see our Privacy Policy

Reviewer #1: No

Reviewer #2: **Yes:** Heng-Chang Chen

Reviewer #3: No

---

## [Editor Report · Acceptance letter]

PONE-D-25-45745R2

PLOS One

Dear Dr. ELAHE,

I'm pleased to inform you that your manuscript has been deemed suitable for publication in PLOS One. Congratulations! Your manuscript is now being handed over to our production team.

Kind regards,

on behalf of

Dr. Tao Huang

Academic Editor

PLOS One